# Suction Leap-Hand: Suction Cups on a Multi-fingered Hand Enables Embodied Dexterity and In-Hand Teleoperation

**Abstract:** This is a technique report to introduce our designed hand with suction cups on its fingertips and palm, Suction Leap Hand (S-Leap Hand). By simply mounting suction cups on a three-fingered dexterous hand, we bring two advantages: much more dexterity for in-hand manipulation and the ability to teleoperate of in-hand manipulation, which are two features that are challenging for existing dexterous hands. We show some demonstrations of in-hand manipulations, e.g., in-hand reorientation, multiple object grasping, and grasping pose changing, that are challenging for current robot hands. Furthermore, we show some tasks that are even challenging for humans, e.g., in-hand peg-in-hole. CAD models and teleoperation codes will be available soon [1].

## 1 Introduction

Dexterous multi-fingered hands are widely studied to enable robots human-like manipulation ability. Dexterous in-hand manipulations are also widely studied, including rigid object manipulation, like object grasping [1], reorientation [2, 3], solving a Rubis's cube [4], multiple object grasping [5], catching [6], and so on.

However, two challenges exist. The first is that current robot hands that even have comparable numbers of joints and degree of freedoms are still far less dexterous than human. There are indeed several cases where using a Shadow Hand or an Allegro Hand performing various in-hand dexterous manipulation, including object reorientation, solving Rubik's cube, and multiple-object grasping, but we wonder, if the hand can be even more dexterous to approaching human-level dexterity in a relatively low cost.

The second challenge is that it is extremely challenging to perform teleoperations of in-hand manipulation for human operator. Considering doing Sim-to-Real transferring is extremely challenging, there are also works focusing on imitation learning by collecting real-world demonstrations without the need of painful Sim-to-Real for gripper manipulation. Is there an easier yet effective solution to teleoperate a dexterous hand for in-hand manipulation?

In this report, our aim is to solve both tasks, including:

- A new multi-fingered hand that can perform human-level in-hand manipulation.
- A teleoperation system that an operator can easily teleoperate the hand to perform dexterous in-hand manipulation.

We aim to offer a simple method for researchers to adapt their Leap-Hand [7] easily with little cost (less than $400). Although our design is based on the popular open-source Leap-Hand design, it

---

[1]Due to double-blind review, we will reveal our project page and the detailed open source content after the review.

Submitted to the 8th Conference on Robot Learning (CoRL 2024). Do not distribute.

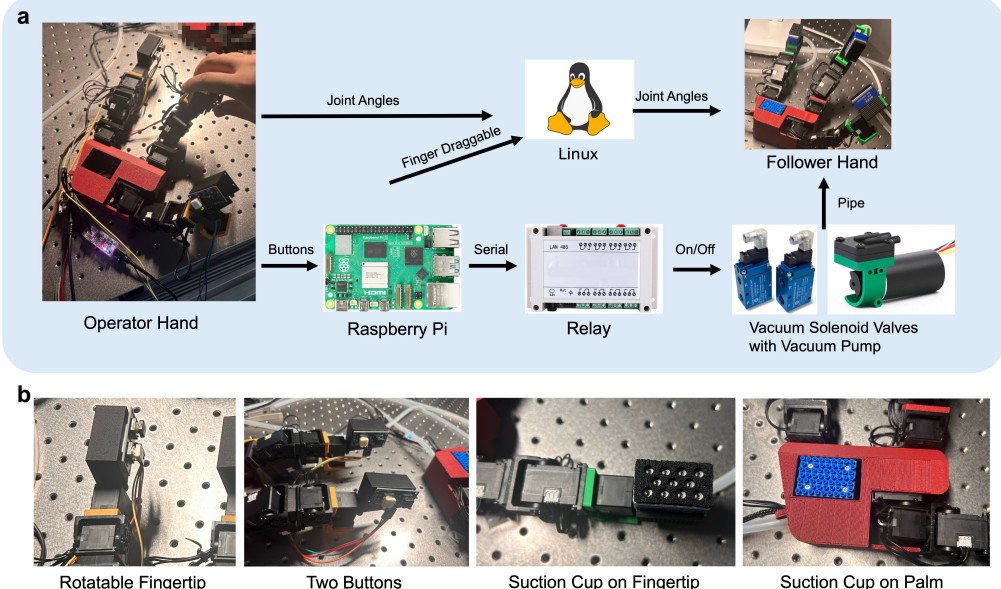

Figure 1: S-Leap Hand design. a. The system contains two hands, one is the operator hand which is dragged by the human operator, and it passes joint angles to the host. When the joint drag-gable button is pressed for once, the finger becomes drag-gable and the follower hand executes the joint angles. The button pressed states are passed to a Raspberry Pi (which can be replaced by an Arduino for cheaper price), further to the relay, and the relay controls the vacuum valves' on/off states to activate or deactivate the suction cups. b. The fingertip is rotatable as the fifth joint on each finger, two buttons are mounted on the operator hand's each fingertip. Sponge-covered suction cups are mounted on each fingertip and the mini suction array based suction cup is on the palm.

is not limited to this specific hand design, and the principle for the hand modification is 1. more DoFs for each finger, 2. fingers are better to be placed far from each other rather than compacted placement, and 3. suction cups mounted on the fingertips and the palm.

## 2 Related Work

**Dexterous hand designs** have been extensively studied since the last century [8], and most of them look alike human hands for easier teleoperation based on hand tracking. Normally, each finger has only four joints or even less because human hands only have four joints for each finger (except the thumb).

**Adhesive end-effector designs** are also not the first time to be proposed. Suction cups are widely used for robotic grasping with the help of the traditional suction cup as the end-effector [9], gecko inspired two-fingered gripper [10], active adhesive materials on a five-fingered hand [11], three fingered submarine gripper [12], suction cups on the back of fingertips of a soft four-fingered hand [13], suction cups on a multi-fingered hand for versatile grasping [14] and so on.

However, these adhesive end-effectors do not fully exploit the manipulation ability beyond grasping. We notice that an effective yet simple solution is to making a single finger more dexterous to make it work as if it is an independent robot arm.

**Teleoperation** of robotic manipulation is crucial to collect real-world training data of high-quality. ALOHA, the dual arm system, has explored how to efficiently build a dual arm system to collect the training data as well as deploying the manipulation policy in the real world [15]. However, this tele-operation becomes extremely difficult for dexterous manipulation with multi-fingered hands. Dex-Gen used a pre-trained policy to understand human's intention during teleopration [16]. DexForce

used human-dragging to register the force during deployment and the force during human-based dragging [17]. Dexterous Cable Manipulation presents a method manually dragging a five-fingered hand simultaneously with two human operators and replaying to collect successfully demonstrations [18]. Moreover, there are exoskeleton devices to help human to collect dexterous demonstrations [19, 20].

## 3 Hand Design

Following Leap Hand design [7], we keep the original thumb, index, and ring fingers layout as well as their joints. The main differences are (see Figure 1):

1. We add one additional joint to the original fingertip, making each finger five joints instead of four. Besides, we remove the middle finger, making each hand three fingers instead of four. This allows an easier way to change a Leap Hand into an S-Leap Hand without purchasing additional servo motors which are the major cost of making a dexterous hand.

2. We mount a small suction cup on each fingertip. We recommend using the rigid suction cup with sponge surface, which allows a wider range of target objects with various surface and material compared to traditional silicone-based round suction cup.

3. We also mount a large suction cup array on the palm. The suction cup array has many mini suction cups on it and each of them can activate independently according if the vacuum forms. As for common suction cups, they will easily cause vacuum leakage. This suction cup array can work to attach objects of small sizes without worrying of vacuum leakage.

## 4 Teleopration Design

Following ALOHA, we have two hands in total, one is the operator hand which needs a human operator, and another is the follower hand which executes the actions by the operator, shown in Figure 1. There are two buttons attached to each fingertip of the operator hand. When pressed the white button, the finger becomes draggable according to the human operator's motions. Once pressed the black button, the suction cup on the fingertip activates and will deactivate after another press. These buttons are connected to a Raspberry PI and communicate with the PC via ROS2.

## 5 Manipulations

We first discuss why mounting suction cups on fingertips enables much more dexterity that is not limited to grasping but a broad range of in-hand manipulations, then we show that our hand can perform many challenging in-hand manipulations, and some are considered to be impossible for current multi-fingered hands.

### 5.1 Suction Cup Helps Improving Dexterity

Traditional contact-based grasping needs at least two fingers to grasp an object (including the palm which can be regarded as a virtual finger [21]). Besides, we also need to ensure the contact is stable by considering the friction cone [22]. The nature of requiring at least two contact areas (two fingers) is that, contact force can only offer the force that is away from the contact area, to compensate the gravity of an object in any pose, two contact forces are required and normally there will be three forces (but mostly fingers are soft and deformable to some extend, so two fingers offer more than two contact points, which are usually contact areas).

Instead, forces provided by suction cups can be in more directions, which is adhesive. One suction cup can basically grasp most objects as long as they at least have one small area for suction cup attaching [23], which greatly reduce the effort on considering how to place two fingers to produce a force-closure grasp candidate.

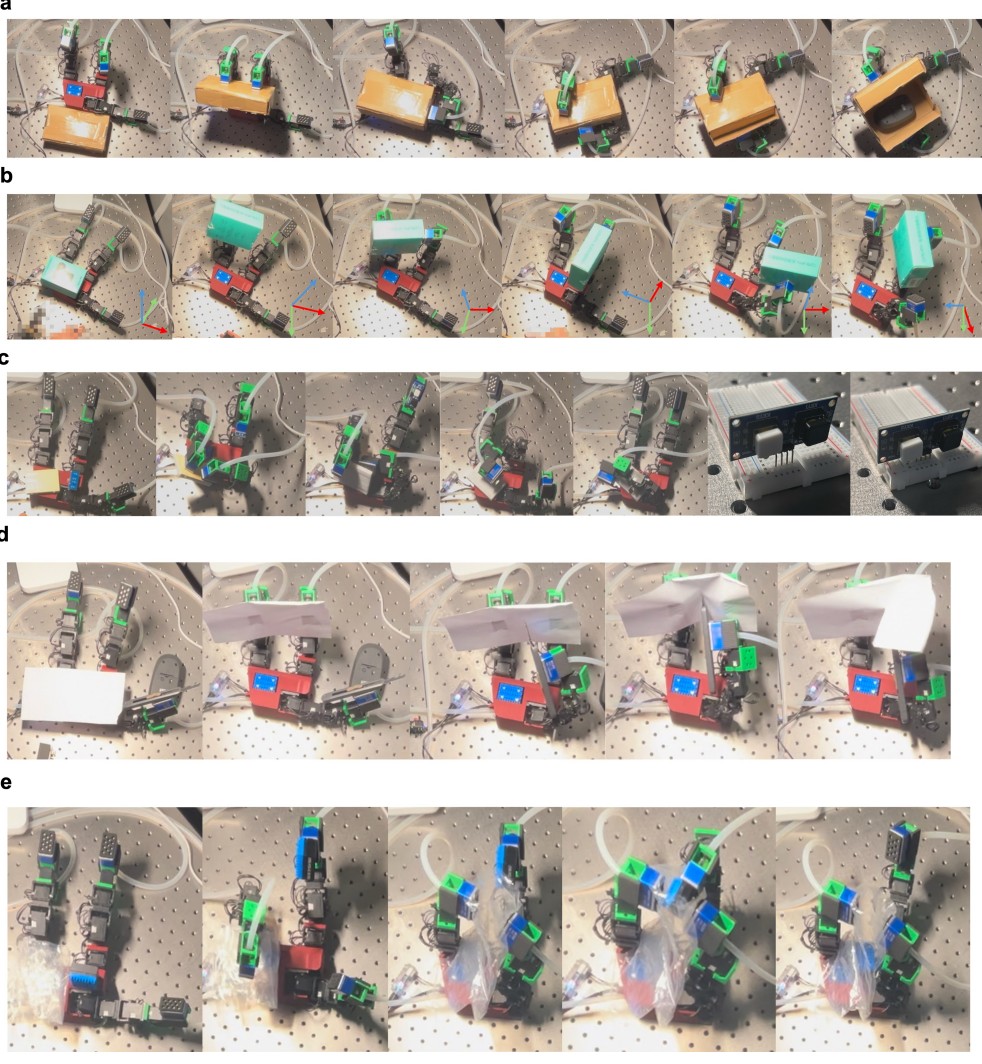

Figure 2: S-Leap Hand performing challenging in-hand manipulation. a. Opening a packed box. b. In-hand re-orientation. c. In-hand breadboard insertion. d. Paper cutting. e. Opening a plastic bag and put an object inside it.

## 5.2 Challenging In-hand Manipulation

In Figure 2, we present five challenging in-hand manipulation by teleoperation. Both hands are fixed on an optical platform and the manipulated object is placed next to or directly on the follower hand. Due to 4-page limitation, we skip many details and refer to our video.

## 6 Conclusion

We present our three-finger S-Leap Hand with suction cups on each rotatable fingertip and the palm, allowing both in-hand manipulation of challenging tasks and teleoperation. We do not focus on using exact algorithm to solve these manipulation tasks, which is not the main topic of this paper. But rather, we show that the hand is capable of doing so, showing a promising manipulation upper limit by teleoperation. With such hardware system, data-driven methods with real-world demonstration of high-quality is possible in the future.

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
