# OpenReview forum: "Suction Leap-Hand: Suction Cups on a Multi-fingered Hand Enables Embodied Dexterity and In-Hand Teleoperation"
_robot-learning.org/CoRL/2025/Workshop/Dexterous_Manipulation — CoRL 2025 Workshop Dexterous Manipulation Spotlight_

### Official Review · Reviewer_Rvv9 · 2025-09-07
**review of Suction Leap-Hand**

**Rating:** 7
**Confidence:** 4

**Review:**

I think this design is interesting. Meanwhile, the suction leap-hand shows good in-hand manipulation and dexterous grasping abilities.

---

### Official Review · Reviewer_sX1p · 2025-09-10
**Review for Suction Leap-Hand**

**Rating:** 6
**Confidence:** 4

**Review:**

The paper introduces the Suction Leap Hand (S-Leap Hand), a low-cost modification of the Leap Hand that integrates suction cups on each fingertip and the palm to enhance dexterity and enable in-hand teleoperation. By adding a fifth joint to each finger and using rigid sponge-surface suction cups, the design allows single-finger grasps and versatile in-hand manipulations that are difficult for existing robot hands, such as peg-in-hole and multi-object reorientation. The authors also present a teleoperation system via a simple operator–follower setup with fingertip buttons and show some teleop results. However, suction-based grasping introduces latency — the activation appears pulsed rather than seamless — and the added hardware makes the hand bulkier. This may reduce the inherent dexterity from the original Leap Hand and limiting its application scenarios.

---

### Decision · Program_Chairs · 2025-09-18

Accept (Spotlight)